# The Association between Post-Traumatic Stress Disorder and Problematic Alcohol and Cannabis Use in a Multi-Ethnic Cohort in The Netherlands: The HELIUS Study

**DOI:** 10.3390/ijerph21101345

**Published:** 2024-10-11

**Authors:** Christin Kühner, Joanne P. Will, Sera A. Lortye, Henrike Galenkamp, Anja Lok, Mirjam van Zuiden, Arnoud R. Arntz, Kathleen Thomaes, Anna E. Goudriaan, Marleen M. de Waal

**Affiliations:** 1Arkin Mental Health Care, Research Department, Amsterdam Institute for Addiction Research, 1033 NN Amsterdam, The Netherlands; joannewill90@gmail.com (J.P.W.); sera.lortye@jellinek.nl (S.A.L.); a.e.goudriaan@amsterdamumc.nl (A.E.G.); marleen.de.waal@arkin.nl (M.M.d.W.); 2Department of Psychiatry, Amsterdam University Medical Centers, University of Amsterdam, 1105 AZ Amsterdam, The Netherlands; a.lok@amsterdamumc.nl (A.L.); kathleen.thomaes@sinaicentrum.nl (K.T.); 3Sinai Centrum, Arkin Mental Health Care, 1186 AM Amstelveen, The Netherlands; 4Department of Public and Occupation Health, Amsterdam University Medical Center, Location AMC, University of Amsterdam, 1105 AZ Amsterdam, The Netherlands; heliuscoordinator@amsterdamumc.nl; 5Center for Urban Mental Health, University of Amsterdam, 1098 XH Amsterdam, The Netherlands; 6Department of Clinical Psychology, Utrecht University, 3584 CS Utrecht, The Netherlands; m.vanzuiden@uu.nl; 7Department of Clinical Psychology, University of Amsterdam, 1018 WS Amsterdam, The Netherlands; a.r.arntz@uva.nl

**Keywords:** post-traumatic stress disorder (PTSD), alcohol use, cannabis use, ethnic minority, Healthy Life in an Urban Setting Study

## Abstract

(1) Background: Ethnic minorities exhibit a higher prevalence of post-traumatic stress disorder (PTSD), while results for problematic substance use among ethnic groups remain mixed. PTSD and problematic substance use often co-occur; however, the impact of ethnicity on this association has not yet been investigated. (2) Methods: Self-report data on problematic alcohol/cannabis use (AUDIT/CUDIT) and presence of severe PTSD symptoms (PSS-SR) of *N* = 22,841 participants of Dutch (*n* = 4610), South-Asian Surinamese (*n* = 3306), African Surinamese (*n* = 4349), Ghanaian (*n* = 2389), Turkish (*n* = 3947), and Moroccan (*n* = 4240) origin were available from the HELIUS study. (3) Results: We found a positive association between the presence of severe PTSD symptoms and problematic alcohol and cannabis use. Ethnicity did not moderate the association between the presence of severe PTSD symptoms and problematic alcohol/cannabis use. (4) Conclusions: We demonstrated the relationship between the presence of severe PTSD symptoms and problematic alcohol/cannabis use in a multi-ethnic sample. The relationship between the presence of severe PTSD symptoms and problematic alcohol/cannabis use was similar between ethnic groups. We recommend screening for PTSD symptoms in those exhibiting problematic substance use and vice versa, regardless of ethnic background.

## 1. Introduction

A recurrent problem in mental health research is the lack of ethnic diversity in study samples. There is a consensus that psychological science currently relies too much on samples from WEIRD (western, educated, industrialized, rich, and democratic) societies, limiting the generalizability of findings [1]. A limited amount of research has focused on ethnic minorities, while populations around the globe are becoming more ethnically diverse [2,3]. Importantly, ethnic minorities have been demonstrated to be at greater risk for both somatic and mental disorders, underlining the relevance of including these groups in scientific research [3]. A recent systematic review has demonstrated elevated rates of anxiety, depression, and post-traumatic stress disorder (PTSD) in European immigrants [4]. In line with this, Blackmore et al. [5] provided meta-analytic evidence that migrants experience PTSD at rates well above the global estimates (31.46% versus 3.90%). A study by Andersson et al. [6] showed that 58% of immigrants in Sweden suffer from PTSD. These rates are greatly exacerbated when compared with prevalence rates of PTSD in Europe, which range from 0.56 to 6.67% [7]. Zooming in on various ethnic groups in the Netherlands, van Leijden et al. [8] utilized a population-based sample to demonstrate that groups of South-Asian Surinamese, African Surinamese, Ghanaian, Turkish, and Moroccan origin all reported higher PTSD symptoms when compared with the Dutch origin group. Potential explanations for the higher rates of PTSD in immigrant groups may be a greater exposure to potentially traumatic events (pre- and post-migration), higher trauma load, discrimination, being exposed to war-related violence, female gender, and lower socio-economic status, as all have been associated with an increased risk for PTSD in previous studies [9,10,11]. Additionally, higher levels of stigma around help-seeking and a language barrier when seeking help may prolong the duration of symptoms, adding to the burden of PTSD [4,9,12].

Another disorder that has been examined in ethnic minorities (migrant/refugee samples) is substance use disorder (SUD), which refers to the persistent use of substances despite negative consequences and loss of control over the amount and frequency of substance use [13]. According to the World Health Organization (WHO), no amount of alcohol/cannabis use is healthy. Most studies on SUD/problematic substance use and ethnicity are conducted in immigrant/refugee samples that contain mostly non-western immigrants. According to a recent global meta-analysis, there are mixed findings in migrant samples, with rates of problematic alcohol use ranging from 4 to 36% and rates of SUD ranging from 1 to 20% [14]. In nationwide cohort studies in Sweden and Finland, migrants and refugees showed consistently lower rates of alcohol-use disorder (AUD), cannabis-use disorder (CUD), and other mental disorders [15,16]. At the same time, a review by the WHO focused on migrants/refugees in Europe showed similar rates of problematic substance use when comparing migrants/refugees to their host populations [17]. A recent population-based cohort study in the Netherlands demonstrated that all ethnic minority groups, except those of Moroccan origin, exhibited lower rates of regular and excessive alcohol consumption, as well as a lower risk for developing alcohol dependence when compared to the Dutch origin group [18]. The authors of this study suggested that this specific risk for problematic alcohol use may be due to religious norms surrounding alcohol consumption in Muslim majority countries (MMCs) such as Morocco. This suggestion is in line with evidence demonstrating that among those who consume alcohol, religious beliefs do not lower the risk of AUD [19]. Furthermore, the stress resulting from non-conformity with religious norms may exacerbate some of the risk factors for problematic alcohol consumption, such as isolation and stigmatization [20]. Another study revealed that although members of ethnic minorities reported less overall use of cannabis, they experienced greater problems associated with using cannabis when compared to the Dutch origin group [21]. Most ethnic minorities may be less frequent users and, at the same time, experience more problems when using alcohol or cannabis. Risk factors for substance use in ethnic minorities are lower socio-economic status, younger age, and male gender [10].

A well-established finding is that PTSD and SUD often co-occur in the population, with 25% of those with SUD concurrently meeting diagnostic criteria for PTSD and 30–60% meeting the diagnostic criteria for PTSD at some point in their lives [22,23]. Vice versa, of those with PTSD, 35% also meet the diagnostic criteria for SUD, and 36–52% meet the diagnostic criteria for AUD [24]. Moreover, patients suffering from both PTSD and SUD show a more severe clinical profile than those with either disorder alone, with worse physical health, more interpersonal problems, and greater severity of substance use [25]. Despite the high prevalence of the comorbidity of PTSD and SUD and the resulting increase in clinical severity, no study to date has examined the co-occurrence of these disorders in European ethnic minorities with specific instruments for these disorders.

In sum, research has demonstrated high estimates of PTSD and mixed findings regarding estimates of SUD in ethnic minorities/migrants in Europe, as well as greater severity of the clinical presentation of comorbid PTSD/SUD. The aim of this study was to examine the association between the presence of severe PTSD symptoms and problematic alcohol and cannabis use across various ethnic groups. To this end, we utilized the population-based cohort, in which others have demonstrated that ethnic minorities scored higher than the Dutch origin group on the measures of the presence of severe PTSD symptoms and problematic cannabis use [8,18,21]. Moreover, they demonstrated that all but the Moroccan origin group scored lower than the Dutch origin group on the measure of problematic alcohol use. Based on previous findings, we expected that (1) the presence of severe PTSD symptoms was positively associated with problematic alcohol and cannabis use [23,26]. We had no a priori expectations whether the relationship between the presence of severe PTSD symptoms and problematic alcohol/cannabis use differs per ethnic group (as compared to the Dutch group).

## 2. Materials and Methods

### 2.1. The HELIUS Study

We used baseline questionnaire data from the *Healthy Life in an Urban Setting (HELIUS)* study, a multi-ethnic prospective cohort study that is conducted in Amsterdam, the Netherlands. Participants between the ages of 18 and 70 were randomly sampled, stratified by ethnicity, from the municipal registry. The baseline data collection took place between 2011 and 2015 in participants with Dutch, Surinamese, Ghanaian, Turkish and Moroccan origin. The protocol of the study has been described extensively elsewhere [3,27]. The HELIUS study is carried out in accordance with the Declaration of Helsinki and has been approved by the AMC Ethical Review Board (MREC 10/100# 17.10.1729). All personal data collected within the study are protected by following general data protection regulation (GDPR) principles [28]. All participants signed written informed consent.

Participants chose to fill in the questionnaires on paper or online and they completed the questionnaires in Dutch, English (Ghanaians), or Turkish. Those who were unable to fill out the questionnaires received help from a trained and ethnically matched interviewer.

### 2.2. PTSD Symptoms

We used the PTSD symptom scale—self-report version (PSS-SR) to assess the presence of severe PTSD symptoms during the preceding two weeks [29]. The PSS-SR consists of nine items: three re-experiencing items, two avoidance items, and four hyperarousal items. Each item corresponds to one of the DSM-IV diagnostic criteria for PTSD. Questions are answered dichotomously (yes/no). A sum score of 7 or higher is considered as experiencing ‘severe PTSD symptoms’ [8]. In the current study, we dichotomized the sum score of the PSS-SR such that participants with a score of zero to six were categorized as ‘0—not severe’ and participants with a score of seven to nine as ‘1—severe’ [8]. Cronbach’s alpha for this questionnaire was .90 in the total sample. The PSS-SR has shown good psychometric properties [8,29].

### 2.3. Problematic Substance Use

We used the alcohol-use disorder identification test (AUDIT) to assess problematic alcohol use during the past twelve months [30]. The AUDIT consists of ten items that are answered on a 5-point Likert scale from zero to four, with higher scores indicating a higher frequency of alcohol consumption and more problems associated with alcohol consumption. The overall score is a sum score across the ten items. The cut-off score for problematic alcohol use is ≥8 [30]. The AUDIT has demonstrated good psychometric properties [30,31], with a Cronbach’s alpha of .78 in the current study. While some studies employed different cut-off scores for men and women, previous research in this sample demonstrated that this leads to the same results [31]. Therefore, we employed the cut-off score of ≥8 for problematic alcohol use in the whole sample. We dichotomized the AUDIT score, such that participants with a score between zero and seven received the label ‘no problematic alcohol use’, and those with a score of eight or higher received the label ‘problematic alcohol use’ [31]. The range of the observed AUDIT scores in the current sample was 0–39.

We used the cannabis-use disorder identification test (CUDIT) to assess problematic cannabis use in the past twelve months [32]. The CUDIT consists of ten items that are answered on a 5-point Likert scale ranging from zero to four, with higher scores indicating a higher frequency of cannabis consumption and more problems associated with cannabis consumption. The overall score is a sum score across the ten items. The cut-off score for problematic cannabis use is 8. The CUDIT has demonstrated good psychometric properties in the past [32], with a Cronbach’s alpha of .80 in the current study. However, a previous study on the HELIUS dataset has demonstrated differential item functioning of the CUDIT across males in the ethnic groups, suggesting that those of non-Dutch origin exhibit a tendency to underreport/overreport on certain items of the CUDIT [21]. Specifically, compared to the Dutch participants, African- and South-Asian Surinamese male participants overreported their problematic cannabis use, while Moroccan and Turkish male participants underreported their problematic cannabis use. We dichotomized the CUDIT score, such that participants with a score between zero and seven received the label ‘no problematic cannabis use’ and those with a score of eight or higher received the label ‘problematic cannabis use’. The range of the observed CUDIT scores in the current sample was 0–40.

### 2.4. Ethnicity

Ethnicity was determined based on the registered country of birth of the participant and of his/her parents, which is presently the most widely accepted definition of ethnicity in the Netherlands [33]. A participant was considered to be of Dutch ethnicity if the person as well as both parents were born in the Netherlands. A participant was considered of non-Dutch ethnicity if either one or two of the following criteria is met: 1. born outside the Netherlands and at least one parent was born abroad (first generation); 2. born in the Netherlands but both parents are born outside the Netherlands (second generation). After data collection, participants of Surinamese ethnic origin were further classified according to self-reported ethnic origin (obtained by questionnaire) into ‘African’, ‘South-Asian’ or ‘Other’. The Dutch origin group served as the reference group.

### 2.5. Covariates

The sociodemographic variables included in the analyses were as follows: age, sex, and educational background. Educational background was defined as the highest qualification obtained in the Netherlands or in the country of birth and consisted of four categories: (1) no education or elementary education only, (2) lower vocational or general secondary education, (3) intermediate vocational or higher secondary education, and (4) higher vocational education or university.

### 2.6. Statistical Analyses

All statistical analyses were conducted in IBM SPSS, version 29. For the description of the sample, we have chosen to report the means (*M*), standard deviations (*SD*), and the percentage of people that scored above the cut-off for instruments assessing the presence of severe PTSD symptoms, problematic alcohol, and cannabis use. In doing so, we hope to enable other researchers to compare their sample to ours. In testing demographic differences between groups, we employed χ^2^ tests. To answer our research questions, we conducted a series of logistic regression analyses, given that our outcomes were binary (score above/below cut off for problematic alcohol/cannabis use). We started by predicting problematic alcohol use (dependent variable) based on the dichotomous measure of the presence of severe PTSD symptoms (independent variable), controlling for age, sex, educational background, and ethnicity (covariates). We proceeded in the second step with the moderation analysis by adding the interaction term of ethnicity and the presence of severe PTSD symptoms to the model. We repeated the same procedure to predict problematic cannabis use (dependent variable) based on the presence of severe PTSD symptoms (independent variable), controlling for age, sex, educational background, and ethnicity (covariates). In all analyses, the Dutch origin group served as the reference group. Results are reported as odds ratios (ORs) with corresponding confidence intervals. Model fit is indicated with a Hosmer–Lemeshow test. We tested two-sidedly, with a predetermined alpha of 0.05.

## 3. Results

In total, 24,789 people took part in the HELIUS study, of which *N* = 23,936 completed the questionnaires. We excluded participants because of (a) missing data on the measures of problematic alcohol use (AUDIT) and/or problematic cannabis use (CUDIT) (*n* = 568), presence of severe PTSD symptoms (PSS-SR; *n* = 523); and/or (b) belonging to one of the ethnic groups that were excluded due to their small size (*n* = 586). The ethnic groups that were excluded due to their small sample size were Javanese Surinamese participants (*n* = 250), Surinamese participants (*n* = 286), and participants with an unknown/other ethnic origin (*n* = 50). The final sample therefore consisted of *n* = 22,841 participants (*n* = 13,119 females, see Table 1). The ethnic groups included in the final sample were as follows: Dutch (*n* = 4610), South-Asian Surinamese (*n* = 3306), African Surinamese (*n* = 4349), Ghanaian (*n* = 2389), Turkish (*n* = 3947), and Moroccan (*n* = 4240).

Table 1 shows an overview of the demographics, both for the entire sample (*n* = 22,841) and stratified per ethnic group. The sample consisted of a majority (57.4%; *n* = 13,119) of women. Participants were on average 43.7 (*SD* = 13.4) years old, with the Moroccan origin group being the youngest and the African-Surinamese origin group being the oldest. The Dutch origin group had the smallest percentage of people with the lowest level of educational background (3.3%) and the largest percentage of people with the highest level of educational background (60.5%). We replicated a significant difference in the presence of severe of PTSD symptoms between the groups (χ^2^ (5) = 434.92, *p* < .001) [8], with the highest rate of severe PTSD in the Turkish (11.3%) and the lowest rate in the Dutch origin group (2.5%). We also replicated a significant difference in problematic alcohol use between the groups (χ^2^ (5) = 182.89, *p* < .001) [18], with the highest rate of problematic alcohol use in the Dutch (25.4%) and the lowest rate in the Moroccan origin group (2.4%). We found a significant difference in problematic cannabis use (χ^2^ (5) = 171.17, *p* < .001), with the highest rate of problematic use in the African Surinamese (5.4%) and the lowest rate in the Ghanaian origin group (1.1%).

### 3.1. Association between the Presence of Severe PTSD Symptoms and Problematic Alcohol Use

Table 2 shows the results of the logistic regression analysis in which we predicted problematic alcohol use based on the presence of severe PTSD symptoms while controlling for age, sex, educational background, and ethnicity (model 1). The model was statistically significant (χ^2^ (10) = 2521.06, *p* < .001) and explained 22.5% (Nagelkerke *R*^2^) of the observed variance in problematic alcohol use. The model correctly classified 90.6% of all participants. However, according to the Hosmer–Lemeshow test, the model did not fit the data well (χ^2^ (8) = 52.89, *p* < .001). Those experiencing severe PTSD symptoms had more than two times the odds of exhibiting problematic alcohol use than those not experiencing severe PTSD symptoms (OR = 2.26, CI_95%_ [1.87, 2.73]). Moreover, belonging to any of the non-Dutch ethnic groups was associated with a lower risk of exhibiting problematic alcohol use, when compared to the Dutch origin group, with OR’s ranging from 0.06 to 0.23. See Table 2 for an overview of the OR’s per ethnic group. Additionally, men had more than four times the odds of exhibiting problematic alcohol use than women (OR = 4.11, CI_95%_ [3.71, 4.56]). In terms of the age of the participants, each additional year was associated with a decrease in the chances of exhibiting problematic alcohol use (OR = 0.98, CI_95%_ [0.98, 0.98]. Intermediate vocational schooling (OR = 0.88, CI_95%_ [0.77, 1.00]) and higher vocational schooling/university (OR = 1.06, CI_95%_ [0.93, 1.20]) were both not associated with a different risk of exhibiting problematic alcohol use when compared to no schooling/lower vocational schooling.

### 3.2. Association between the Presence of Severe PTSD Symptoms and Problematic Alcohol Use across Ethnic Groups

We examined whether ethnicity moderates the relationship between the presence of severe PTSD symptoms and problematic alcohol use while controlling for age, sex, and educational background. Table 2 shows the results of adding the interaction term of severe PTSD and ethnicity to the logistic regression model described above. We employed a hierarchical logistic regression analysis, which allowed us to examine the difference in explained variance between step 1 and step 2, as well as the unique contribution of each predictor in the model. The overall model was still statistically significant (χ^2^ (15) = 2524.90, *p* < .001) and explained 22.5% (Nagelkerke *R*^2^) of the observed variance in problematic alcohol use. The model correctly classified 90.6% of all participants. At the same time, adding the interaction effects did not improve the model fit (χ^2^ (5) = 3.88, *p* = .57), which was still poor according to the Hosmer–Lemeshow test (χ^2^ (8) = 52.75, *p* < .001). Therefore, the maximum model fit that we could reach with the variables in our model was already reached before adding the interaction effects. This is in line with the non-significant *p*-values for all PTSD*Ethnicity interaction effects, with ORs ranging from 0.95 to 1.57; for an overview, see Table 2. Therefore, ethnicity did not moderate the relationship between severe PTSD and problematic alcohol use.

### 3.3. Association between Severe PTSD Symptoms and Problematic Cannabis Use

Table 3 shows the results of the logistic regression analysis in which we predicted problematic cannabis use based on the presence of severe PTSD symptoms while controlling for age, sex, educational background, and ethnicity. The model was significant (χ^2^ (10) = 782.96, *p* < .001) and explained around 15.0% (Nagelkerke *R*^2^) of the variance of problematic cannabis use. The model correctly classified 97.2% of cases and showed a good fit for the data according to the Hosmer–Lemeshow test (χ^2^ (8) = 10.86, *p* = .21). Those experiencing severe PTSD symptoms had almost three times the odds of exhibiting problematic cannabis use, compared to those not experiencing severe PTSD symptoms (OR = 2.96, CI_95%_ [2.31, 3.79]). Moreover, belonging to the African-Surinamese origin group was associated with a significantly higher risk of exhibiting problematic cannabis use, when compared to the Dutch origin group (OR = 2.37, CI_95%_ [1.81, 3.09]). Belonging to either the Turkish (OR = 0.46, CI_95%_ [0.33, 0.65]) or Moroccan (OR = 0.69, CI_95%_ [0.51, 0.94]) origin group was associated with a lower risk of exhibiting problematic cannabis use when compared to the Dutch origin group. Belonging to the South-Asian Surinamese origin group was associated with a similar risk for problematic cannabis use when compared with the Dutch origin group (OR = 1.11, CI_95%_ [0.83, 1.50]). Additionally, men had more than five times the odds of exhibiting problematic cannabis use, when compared to women (OR = 5.26, CI_95%_ [4.34, 6.36]). In terms of age of the participants, each additional year was associated with a decrease in the chances of exhibiting problematic cannabis use (OR = 0.96, CI_95%_ [0.95, 0.96]). Intermediate vocational schooling (OR = 0.70, CI_95%_ [0.58, 0.84]) and higher vocational schooling/university (OR = 0.33, CI_95%_ [0.26, 0.43]) were both associated with a significant decrease in the risk of exhibiting problematic cannabis use when compared to no schooling/lower vocational schooling. See Table 3 for all test-statistics.

### 3.4. Associations between the Presence of Severe PTSD Symptoms and Problematic Cannabis Use across Ethnic Groups

We examined whether ethnicity moderates the relationship between the presence of severe PTSD symptoms and problematic cannabis use while controlling for age, sex, educational background, and ethnicity. Table 3 shows the results of adding the interaction term of the presence of severe PTSD symptoms and ethnicity to the logistic regression model described above. However, the group of those of Ghanaian origin who reported problematic cannabis use and who scored above the cut-off for the presence of severe PTSD symptoms was too small to compute the interaction effect and thus be included in the moderation analysis (see Table 1 for *N* and % that scored above the cut-off [8]). We employed a hierarchical logistic regression analysis, which allowed us to examine the difference in explained variance between step 1 and step 2, as well as the unique contribution of each predictor in the model. The overall model was still statistically significant (χ^2^ (15) = 790.92, *p* < .001) and explained 15.1% (Nagelkerke *R*^2^) of the observed variance in problematic cannabis use. The model correctly classified 97.2% of all participants. Adding the interaction effects did not improve the model (χ^2^ (5) = 7.96, *p* = .16), which still fit the data well as indicated by the Hosmer–Lemeshow test (χ^2^ (8) = 12.83, *p* = .12). None of the interaction effects of PTSD*Ethnicity were significant, with ORs ranging from 0.44 to 0.61; for an overview, see Table 3. Therefore, ethnicity did not moderate the relationship between the presence of severe PTSD and problematic cannabis use.

## 4. Discussion

In the current study, we investigated the association between the presence of severe PTSD symptoms and problematic alcohol/cannabis use in a large multi-ethnic cohort in the Netherlands. Moreover, we investigated whether ethnicity moderates the relationship between the presence of severe PTSD symptoms and problematic alcohol/cannabis use. Experiencing severe PTSD symptoms increased the odds, while being of non-Dutch origin decreased the odds of exhibiting problematic alcohol use. Ethnicity did not moderate the relationship between severe PTSD symptoms and problematic alcohol use. Experiencing severe PTSD symptoms and belonging to the African-Surinamese origin group increased the odds of exhibiting problematic cannabis use. Belonging to the Turkish- or Moroccan-origin group decreased the odds of exhibiting problematic cannabis use. Ethnicity did not moderate the relationship between severe PTSD symptoms and problematic cannabis use.

Our finding that the presence of severe PTSD symptoms and problematic alcohol use are related is in line with a large body of literature [22,23,24]. Importantly, we demonstrated the lower rates of problematic alcohol use in ethnic minorities, while accounting for the fact that these groups experience much higher rates of severe PTSD [17]. Moreover, we demonstrated the robustness of the relationship between the presence of severe PTSD symptoms and problematic alcohol use in a multi-ethnic sample, expanding the scope of this finding to non-WEIRD samples. A potential explanation for the lower rates of problematic alcohol consumption in the non-Dutch origin groups may be the religious rules surrounding alcohol consumption in MMCs, such as Morocco or Turkey [34]. At the same time, the relationship between the presence of severe PTSD symptoms and problematic alcohol use was similar in each ethnic group when compared to the Dutch origin group, as indicated by the non-significant moderation analysis. This finding is especially interesting given the differences in prevalence of the presence of severe PTSD symptoms and problematic alcohol use between ethnic groups. Based on this exploratory study, it may be that the mechanisms underlying the presence of severe PTSD symptoms and problematic alcohol use differ in ethnic groups, as indicated by their different prevalences. However, mechanisms that underly the *co-occurrence* of the presence of severe PTSD symptoms and problematic alcohol use may not differ between ethnic groups. Future research should investigate potential shared and different mechanisms in majority non-Caucasian samples to disentangle the factors contributing to the occurrence of PTSD/SUD/ co-occurring PTSD and SUD in different ethnic groups.

Our finding that severe PTSD symptoms and problematic cannabis use are related is in line with studies in psychiatric outpatients [35]. We demonstrated the relationship in a multi-ethnic sample, broadening the scope of our results to non-WEIRD and population-based samples. Importantly, we expanded upon previous studies by demonstrating the effect of ethnicity on problematic cannabis consumption while controlling for the presence of severe PTSD symptoms in ethnic minorities [17]. Our findings are in line with the WHO-initiated review and cohort studies in Sweden and Finland whereby ethnic minorities showed similar or lower rates of problematic substance use when compared to the host population [15,16,17]. However, in our sample, the South-Asian Surinamese-origin group showed a higher rate of problematic substance use than the Dutch-origin group. A potential explanation for the lower rates of problematic cannabis consumption in (all but one) ethnic minorities are religious rules surrounding substance use [34]. Under Dutch law, both alcohol and cannabis consumption are legal and therefore potentially less stigmatized. If one comes from a cultural background where either or both of those substances (alcohol/cannabis) are illegal by (religious) law or stigma associated with consumption (e.g., MMC), then underreporting the use of these substances may be more likely. Interestingly, recent meta-analytic evidence suggests that across a variety of countries, problematic alcohol use is less stigmatized than other drug use, such as problematic cannabis use [36].

The relationship between the presence of severe PTSD symptoms and problematic cannabis use was similar in each ethnic group when compared to the Dutch-origin group, as indicated by the non-significant moderation analysis. Also this finding becomes more interesting in the context of the above-mentioned differences in prevalence of the presence of severe PTSD symptoms and problematic cannabis use in the ethnic groups. We propose a similar explanation as for the similarity of the relationship between the presence of severe PTSD symptoms and problematic alcohol use across ethnic groups. It could be that the mechanisms underlying the presence of severe PTSD symptoms and problematic cannabis use differ in ethnic groups, as indicated by the different prevalences in the presence of severe PTSD symptoms and problematic cannabis use. At the same time, the mechanisms that underly the *co-occurrence* of the presence of severe PTSD symptoms and problematic cannabis use may not differ between ethnic groups, as should be investigated in future research. Another potential explanation for our findings concerns the use of culturally sensitive measurement methods. The AUDIT and CUDIT were developed by the WHO and could be answered in the participant’s language of choice. However, as stated earlier, the CUDIT has demonstrated differential item functioning for the male portion of the current sample [21]. Specifically, African- and South-Asian Surinamese male participants over- and Moroccan and Turkish male participants underreported their problematic cannabis use. Moreover, the AUDIT/CUDIT/PSS-SR are based on our Western notion of psychopathology, which may not align with the perception of those with non-Western origins, leading to under- or overreporting on some of the questionnaires [37]. The high rate of problematic cannabis use in the African-Surinamese origin group may be an artifact of overreporting on the CUDIT. Relatedly, the most commonly used cut-off for problematic alcohol and cannabis use is a score bigger than or equal to eight on the AUDIT and CUDIT. Importantly, this is a screening cut-off that indicates that a healthcare professional should look more closely at the frequency of consumption and potential problems associated with it. It does not constitute a diagnosis and is prone to biases inherent to retrospective self-report measurement, such as recall-bias [38]. Moreover, the AUDIT/CUDIT cover a long time-span (twelve months), during which consumption may have fluctuated substantially. At the same time, the PSS-SR covers a much shorter time-span (two weeks), leading to substantial differences in the period during which symptoms are assessed.

### 4.1. Strengths and Limitations

The current study has several strengths, such as the use of an ethnically diverse sample of the population in Amsterdam, with a large sample size and large subsamples of five ethnic minority groups. There are certain limitations in the current study, such as the use of retrospective self-report questionnaires for all constructs under investigation (presence of severe PTSD symptoms, problematic alcohol/cannabis use), which may be tainted by recall bias [39]. Participants could choose among different languages and ask for assistance while filling out the questionnaires, but we cannot exclude the possibility that some participants may have misunderstood items/questionnaires. Previous research demonstrated that the CUDIT showed differential item functioning for some ethnic groups in the current sample. This implies that the probability of some ethnic groups endorsing certain items on the CUDIT differed substantially from the probability of the Dutch group endorsing those items. The current study was cross-sectional, meaning no conclusions can be drawn about the directionality of the relationship between the presence of severe PTSD symptoms and problematic alcohol/cannabis use. Moreover, the AUDIT/CUDIT/PSS-SR are screening instruments and cannot substitute diagnosis by a healthcare professional.

### 4.2. Future Prospects

Future research should investigate the cultural validity of the AUDIT/PSS-SR in ethnic minorities in Europe. Given that the CUDIT has already demonstrated differential item functioning for some ethnic minorities, we recommend the development of a screening tool for problematic cannabis use that is more culturally sensitive [21]. Moreover, we recommend the use of clinical samples to test whether the current findings can be replicated with diagnosed (instead of screened) PTSD/AUD/CUD. Finally, we recommend studies on underlying mechanisms of the link between PTSD and SUD to include non-WEIRD samples and examine potential differences in mechanisms among ethnic groups. Potential mechanisms may be elucidated by a combination of qualitative and quantitative research, whereby qualitative research can give rise to ideas that can be tested using quantitative methods. Moreover, longitudinal research on the relationship between PTSD and SUD may lead to more clarity regarding the directionality of this comorbidity, i.e., whether the PTSD symptoms are a risk factor for or consequence of SUD symptoms.

## 5. Conclusions

We have demonstrated the relationship between the presence of severe PTSD symptoms and problematic alcohol and cannabis use in a multi-ethnic (non-WEIRD) sample, underlining the robustness of this association. We did not find that ethnicity moderated the relationship between the presence of severe PTSD problems and problematic alcohol/cannabis use. Therefore, we recommend screening for PTSD symptoms in those presenting with symptoms of SUD and vice versa, regardless of ethnic background. Given the high rates of the presence of severe PTSD symptoms in all non-Dutch origin groups, we recommend screening for PTSD in primary care settings in those populations, such that interventions may be offered where necessary.

## Figures and Tables

**Table 1 ijerph-21-01345-t001:** Characteristics of the Whole Sample and each Ethnic Group.

	Dutch *n* = 4610	South-Asian Surinamese *n* = 3306	African-Surinamese *n* = 4349	Ghanaian *n* = 2389	Turkish *n* = 3947	Moroccan *n* = 4240	Total Sample *N* = 22,841
**Female *N* (%)**	2490 (54.0 %)	1763 (53.3%)	2592 (59.6%)	1464 (61.3%)	2177 (55.2%)	2633 (62.1%)	13,119 (57.4%)
***M*_age_ (*SD*)**	46.10 (14.04)	45.01 (13.55)	47.49 (12.76)	44.13 (11.57)	39.80 (12.42)	39.71 (13.04)	43.73 (13.39)
**Education**							
**1 (%)**	3.3	14.0	5.6	27.7	30.9	30.1	17.6
**2 (%)**	14.2	33.1	35.8	40.1	25.0	18.0	26.3
**3 (%)**	22.0	30.1	36.1	25.9	29.4	34.5	29.9
**4 (%)**	60.5	22.7	22.5	6.2	14.7	17.4	26.2
**PSS-SR**							
***M* (*SD*)**	0.90 (1.79)	1.71 (2.62)	1.23 (2.19)	0.97 (1.94)	1.89 (2.78)	1.65 (2.69)	1.40 (2.40)
**AUDIT**							
***M* (*SD*)**	5.68 (4.55)	2.45 (4.18)	2.51 (3.51)	1.78 (3.34)	1.10 (3.08)	0.52 (2.47)	2.47 (4.02)
**CUDIT**							
***M* (*SD*)**	0.51 (2.27)	0.74 (3.31)	1.08 (3.61)	0.21 (1.72)	0.40 (2.48)	0.52 (3.01)	0.60 (2.87)
**PSS-SR ≥ 7 *N* (%)**	113 (2.5)	312 (9.4)	214 (4.9)	77 (3.2)	445 (11.3)	430 (10.1)	1.591 (7.0)
**AUDIT ≥ 8 *N* (%)**	1.1170 (25.4)	281 (8.5)	298 (6.9)	142 (5.9)	162 (4.1)	101 (2.4)	2.154 (9.4)
**CUDIT ≥ 8 *N* (%)**	87 (1.9)	115 (3.5)	236 (5.4)	26 (1.1)	75 (1.9)	99 (2.3)	638 (2.8)

Note. Education was coded as follows: (1) no education or elementary education only, (2) lower vocational or general secondary education, (3) intermediate vocational or higher secondary education, and (4) higher vocational education or university. *M* = mean, *SD* = standard deviation, PSS-SR= PTSD symptoms scale—self report version, AUDIT = alcohol-use disorder identification test, CUDIT = cannabis-use disorder identification test. The PSS-SR ≥ 7, AUDIT ≥ 8, and CUDIT ≥ 8 rows indicate the percentage of people that score above the cut-off for severe PTSD symptoms, problematic alcohol use, and problematic cannabis use, respectively.

**Table 2 ijerph-21-01345-t002:** Results of the Logistic Regression Analyses with Problematic Alcohol Use as Outcome.

	OR	LL_95%_	UL_95%_	*p*-Value	OR	LL_95%_	UL_95%_	*p*-Value
		Model 1				Model 2		
**Severe PTSD**	2.26	1.87	2.73	**<.001**	2.08	1.37	3.15	**<.001**
**Male Gender**	4.11	3.71	4.56	**<.001**	4.12	3.72	4.56	**<.001**
**Age**	0.98	0.98	0.98	**<.001**	0.98	0.98	0.98	**<.001**
**Education 3**	0.88	0.77	1.00	.06	0.88	0.77	1.00	.06
**Education 4**	1.06	0.93	1.20	.40	1.06	0.93	1.20	.41
**South-Asian Surinamese**	0.23	0.20	0.27	**<.001**	0.23	0.20	0.27	**<.001**
**African-Surinamese**	0.22	0.19	0.26	**<.001**	0.21	0.18	0.25	**<.001**
**Ghanaian**	0.19	0.15	0.23	**<.001**	0.18	0.15	0.23	**<.001**
**Turkish**	0.10	0.08	0.12	**<.001**	0.10	0.08	0.12	**<.001**
**Moroccan**	0.06	0.05	0.08	**<.001**	0.06	0.05	0.08	**<.001**
**South Asian-Surinamese*PTSD**					0.95	0.54	1.67	.86
**African-Surinamese*PTSD**					1.57	0.87	2.84	.14
**Ghanaian*PTSD**					1.29	0.55	3.01	.56
**Turkish*PTSD**					0.95	0.52	1.75	.88
**Moroccan*PTSD**					1.10	0.56	2.15	.78

Note. Sex was coded such that females = 0 and males = 1. Education 3 refers to intermediate vocational schooling compared to no schooling/lower vocational schooling. Education 4 refers to higher vocational schooling/university compared to no schooling/lower vocational schooling. OR = Odds ratio. The ORs for all ethnic groups were computed in reference to the Dutch group. LL_95%_ and UL_95%_ refer to the lower limit and upper limit of the 95% confidence interval. Step 1 refers to the logistic regression analyses in which we predicted problematic alcohol use based on severe PTSD while controlling for age, sex, educational background, and ethnicity. In step 2, we added the interaction term severe PTSD*Ethnicity (moderation analysis). PTSD symptoms were assessed using the PSS-SR. *p*-values smaller than 0.05 are printed bold.

**Table 3 ijerph-21-01345-t003:** Results of the Logistic Regression Analyses with Problematic Cannabis Use as Outcome.

	OR	LL_95%_	UL_95%_	*p*-Value	OR	LL_95%_	UL_95%_	*p*-Value
		Model 1				Model 2		
**Severe PTSD**	2.96	2.31	3.79	**<.001**	5.67	2.68	11.99	**<.001**
**Gender**	5.26	4.34	6.36	**<.001**	5.24	4.33	6.35	**<.001**
**Age**	0.96	0.95	0.96	**<.001**	0.96	0.95	0.96	**<.001**
**Education 3**	0.70	0.58	0.84	**<.001**	0.70	0.58	0.84	**<.001**
**Education 4**	0.33	0.26	0.43	**<.001**	0.33	0.26	0.43	**<.001**
**South-Asian Surinamese**	1.11	0.83	1.50	.49	1.18	0.86	1.63	.30
**African-Surinamese**	2.37	1.81	3.09	**<.001**	2.52	1.91	3.33	**<.001**
**Ghanaian**	0.37	0.23	0.58	**<.001**	0.42	0.26	0.67	**<.001**
**Turkish**	0.46	0.33	0.65	**<.001**	0.49	0.34	0.70	**<.001**
**Moroccan**	0.69	0.51	0.94	.02	0.70	0.50	0.98	.04
**South-Asian Surinamese*PTSD**					0.48	0.20	1.20	.12
**African-Surinamese*PTSD**					0.44	0.18	1.10	.08
**Ghanaian*PTSD**					n.a.	n.a.	n.a.	n.a.
**Turkish*PTSD**					0.51	0.20	1.30	.16
**Moroccan*PTSD**					0.61	0.25	1.48	.27

Note. Sex was coded such that females = 0 and males =1. Education 3 refers to intermediate vocational schooling compared to no schooling/lower vocational schooling. Education 4 refers to higher vocational schooling/university compared to no schooling/lower vocational schooling. OR = Odds ratio. The ORs for all ethnic groups were computed in reference to the Dutch group. LL_95%_ and UL_95%_ refer to the lower limit and upper limit of the 95% confidence interval. Step 1 refers to the logistic regression analyses in which we predicted problematic cannabis use based on severe PTSD while controlling for age, sex, educational background, and ethnicity. In step 2, we added the interaction term of severe PTSD*Ethnicity (moderation analysis). PTSD symptoms were assessed using the PSS-SR. *p*-values smaller than 0.05 are printed bold.

## Data Availability

The HELIUS data are owned by the Amsterdam University Medical Centers, location AMC in Amsterdam, The Netherlands. Any researcher can request the data by submitting a proposal to the HELIUS Executive Board as outlined at http://www.heliusstudy.nl/en/researchers/collaboration (accessed on 23 July 2024), by email: heliuscoordinator@amsterdamumc.nl. The HELIUS Executive Board will check proposals for compatibility with the general objectives, ethical approvals and informed consent forms of the HELIUS study. There are no other restrictions to obtaining the data and all data requests will be processed in the same manner.

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
