# Peer review of "The Association between Post-Traumatic Stress Disorder and Problematic Alcohol and Cannabis Use in a Multi-Ethnic Cohort in The Netherlands: The HELIUS Study"

_ijerph, 2024, doi:10.3390/ijerph21101345_

Round 1

Reviewer 1 Report

Comments and Suggestions for Authors

Please see the word file attached

Author Response

Dear Reviewer, 

We would like to thank you for the time and effort put into revising our manuscript. We have attached our replies in word-format. 

Reviewer 2 Report

Comments and Suggestions for Authors

I would like to thank for the opportunity to review this manuscript. It elucidates the association between post-traumatic stress disorders and the use of alcohol and cannabis, sounding interesting. Since it can give a real contribution in the current literature, it needs some comments to improve it. 

  1. 1) The manuscript requires an English language editing. Please, pay attention on punctuation signs and singular/plural declination of verbal times. 

  1. 2) In the abstract the acronym PTSD appears for the first time. Please, write the full noun. 

  1. 3) What is the meaning of WEIRD and SUD? Please, write the respective full nouns. 

  1. 4) The introduction should be addressed on post-traumatic stress disorders from both clinical and epidemiological point of view, including the risk factors of their onset. After that, it is possible to explain the lack of studies in ethnic diversity in this issue. Please, revise it. 

  1. 5) The full noun “western, educated, industrialized, rich and democratic” is too long and compromises the readability of the sentence. Please, write the acronym before and insert the following full noun into brackets. 

  1. 6) As I mentioned above, post-traumatic stress disorders could have several causes, not only the migrant conditions. Please, revise it focusing on the causes explored in this study. 

  1. 7) According to the ICD-10 tool (https://icd.who.int/browse10/2019/en), substance use disorder is not included in the mental, but it is the cause, as it is reported “Mental and behavioural disorders due to psychoactive substance use (F10-F19). Please, revise the sentence “Another mental disorder that has been examined in migrant/refugee samples is sub-56 stance use disorder (SUD), […] 

  1. 8) Please, explain better the hypothesized or already detected factors associated with substance use disorder in immigrant population. 

  1. 9) Since the title seems to be focused on Multi-Ethnic Cohort, explain better the reasons to focus on the immigrants, even if it could be logical for the experts in this field. 

  1. 10) Write something about the health impact of alcohol and cannabis use in ethnic minorities affected by post-traumatic stress disorder or not affected (making the difference if it is possible) and explain better the reasons for which this population is more vulnerable than others. 

  1. 11) Considering this sentence “A recent population-based cohort study in the Netherlands demonstrated that all ethnic minority groups, except those of Moroccan origin, exhibited lower rates of regular and excessive alcohol consumption, as well as a lower risk for developing alcohol dependence, when compared to the Dutch origin group, explain better what factors could be associated with the alcohol consumption in these ethnic minority groups, please. The same should be done for the following sentences. 

  1. 12) The study population, in this case the ethnic groups, is a part of materials and methods. Please, insert this part in the appropriate section of the manuscript. Moreover, explain better the reasons for choosing specific ethnic groups and excluding the others. 

  1. 13) The lines 89-90 elucidate the study design. Please, insert this part in the section of materials and methods. 

  1. 14) The lines 90-92 describe your hypothesis. Please, revise and improve this part of the text. 

  1. 15) The lines 96-98 explain the statistical analysis used in this study.

  2. 16) Please, insert this part of the test in the statistical analysis section. 

  1. 17) The study period began in 2011, with last update in 2015. We are in 2024. Please, explain the reasons for publishing this data after too many years (about 9 years) and the theoretical and practical implications of this data in the appropriate section of the manuscript. 

  1. 18) In the materials and methods, it is important to elucidate the research methods, such as inclusion and exclusion criteria. The lines 112-122 report too many details for this section, which are more appropriate in the results section. Please, insert the mentioned lines in the results section and add the inclusion and exclusion criteria of this study in the materials and methods section. 

  1. 19) Explain better how to manage personal data of this population, legislative references and other international guidelines to protect personal data and conduct the study in the ethic way. 

  1. 20) Regarding the lines 135-147, insert the maximum AUDIT score, please. Do the same also for the CUDIT score (lines 148-163) 

  1. In the reference to the lines 183-184, deepen the series of logistic regression analyses used in this study, explain better which are the dependent variables and the independent variables. Moreover, add the confounding variables taken in consideration to adjust the logistic regression. 

  1. 21) Considering the descriptive analysis in the first part of the results section, insert such analysis in the statistical analysis section, please. 

  1. 22) Please, add the minimum sample size of this study. 

  1. 23) Explain the reasons for choosing to use mean and SD for quantitative variables and add such reasons in the statistical analysis section, please. 24) Generally, the choice of using a measure of central tendency (and the related measure of dispersion) depends on the result of a normality test. 

  1. 25) Considering the lines 199-207, insert the statistical analysis used to compare the groups in the appropriate section (statistical analysis), please. Moreover, add the measures of association (odds ratio) and the related confidence intervals in the statistical analysis section. 

  1. 26) In the line 246 and line 297, there is an unnecessary comma. 

  1. 27) Please, insert Table 2 after the paragraph entitled “3.2. Association Between the Presence of Severe PTSD Symptoms and Problematic Alcohol Use 248 Across Ethnic Groups” 

  1. 28) In the reference to the lines 250-254, deepen the moderation analysis and the hierarchical logistic regression analysis in the statistical analysis section, adding the reasons for this choice. Then, the meaning of Nagelkerke R2, considering among articles readers there are not educated professionals in statistics, so it could be difficult for them understanding some statistical details. I do not ask to write a chapter of statistics, but very little explanation of specific measures and regressions used in this study. The same is for the lines 258-263 and 265-272, in which it is mentioned a statistical test not reported in statistical analysis (Hosmer Lemshow test). 

  1. 29) The lines 262-263 are a deduction from the results. Please, add them in the discussion section, deepening them in the light of the current literature. 

  1. 30) Please, write the full noun of LL95% an UL95% in Table 2 and 3. 

  1. 31) Please, insert Table 3 after the paragraph entitled “3.4. Associations Between the Presence of Severe PTSD Symptoms and Problematic Cannabis Use Across Ethnic Groups” 

  1. 32) In the discussion section, it is important to compare your results with the current literature with similar results. Please, improve this aspect. 

  1. In the discussion section, elucidate psychological implications with much more details and explain better the biological mechanisms which lead ethnic minorities affected by post-traumatic stress disorder to an alcohol or cannabis use.  

  1. 33) Explain better the future implications of this study and future directions for new studies or new interventions which can planned in this population. 

  1. 34) Check in the current literature whichever recommendations on alcohol or cannabis use in this population and add in the introduction the results of such literature research.

Comments on the Quality of English Language

Minor editing is required in order to improve the manuscript quality.

Author Response

(The authors gave the same response as above.)

Reviewer 3 Report

Comments and Suggestions for Authors

Kühner et al probed the possible association of problematic alcohol and cannabis use in the Dutch population with a focus on the possible association with specific ethnic minorities (i.e., South-Asian Surinamese, African Surinamese, Ghanaian, Turkish and Moroccan) as compared with the Dutch population which served as the reference group. No difference was observed in the association for either cannabis use or PTSD with ethnic origin. This represents an analysis based on HELIUS study data, a large prospective cohort study aimed at investigating possible determinants of social health in a multiethnic population in the Netherlands. Overall, this research article appears well-written, clearly organised and is informative on a complex subject pertaining to possible determinants of social vulnerability. 

I have no specific concerns regarding the present manuscript, but I am curious about possible differences between first vs second generation non-Dutch participants. Research from different fields has shown possible differences in first- or second-generation for various outcomes in mental health (e.g., suicide prevalence). This piece of information appears available as the study authors indicate first- or second-generation non-Dutch participants as a definition to define ethnicity. However, I did not recall this element added to the analysis. The authors may consider (albeit I consider this a non-binding suggestion) adding a brief subanalysis detailing potential differences in this regard for the studied outcomes. 

Author Response

Dear reviewer, 

We would like to thank you for the time and effort put into revising our manuscript. We have attached our answers in word-format. 

Round 2

Reviewer 2 Report

Comments and Suggestions for Authors

I congratulate with the Authors for the improvement of the manuscript. However, I would like to add some comments to increase its quality.

1. Please, pay attention on the space (line 31);

2. Please, add the acronym of World Health Organization (line 66);

3. Please, remove an unnecessary full stop at the line 81;

4. Please, add the full noun of the acronym GDPR and the related references about personal data management and protection in Europe (the European Regulation);

5. Please, considering the response no. 7 of the rebuttal letter, explain better the reason for choosing DSM-IV diagnostic criteria instead of DSM-5 (line 133);

6. Please, explain better the reason for choosing also the median if you have already calculated the mean. I would suggest the median is removed (lines 191-192);

7. Considering the lines 206-207, add the percentage of the confidence intervals and the crude and adjusted odds ratio (cOR and aOR, respectively) if you calculated, otherwise only the full noun of odds ratio and the following acronym (OR). After the confidence interval, put a full stop and start a new sentence;

8. Please, remove N = because the total number of the participants is understandable without it (line 210);

9. Please, remove the full nouns of the acronyms if you have already written above (line 212);

10. Please, write N = in lower-case letter (line 221);

11. Please, check the space after the full stop at the line 222;

12. Considering the lines 228-231, the comparisons to the results published in the literature must be written in the discussion section, not in the results (which reports only the results obtained in the current study);

13. As mentioned in this list (no. 6), remove the median from the Table 1;

14. Please, justify the text from the page 7 of 16;

15.  Please, add only the acronym of odds ratio if the full noun was written above (line 17 at the page 9 of 16).

Comments on the Quality of English Language

Minor editing of English language is required for improvement of the readability (e.g. the repetition of the current study between the lines 108 and 110 from the page 13 of 16).
